# An Implementation Evaluation of the Smartphone-Enhanced Visual Inspection with Acetic Acid (SEVIA) Program for Cervical Cancer Prevention in Urban and Rural Tanzania

**DOI:** 10.3390/ijerph21070878

**Published:** 2024-07-05

**Authors:** Alyssa L. Ferguson, Erica Erwin, Jessica Sleeth, Nicola Symonds, Sidonie Chard, Safina Yuma, Olola Oneko, Godwin Macheku, Linda Andrews, Nicola West, Melinda Chelva, Ophira Ginsburg, Karen Yeates

**Affiliations:** 1Public Health Agency of Canada, Ottawa, ON K1A 0K9, Canada; fe3@ualberta.ca; 2Department of Medicine, Queen’s University, Kingston, ON K7L 3N6, Canada; 3Canadian Cancer Trials Group, Kingston, ON K7L 2V5, Canada; 4School of Medicine, Queen’s University, Kingston, ON K7L 3N6, Canada; 5Royal College of Surgeons in Ireland, D02 YN77 Dublin, Ireland; 6Ministry of Health, Community Development, Gender, Elderly, and Children, Dodoma 40478, Tanzania; 7Kilimanjaro Christian Medical Center, Moshi P.O. Box 3010, Tanzania; 8Pamoja Tunaweza Women’s Centre, Moshi P.O. Box 8434, Tanzania; 9Center for Global Health, National Cancer Institute, Bethesda, MD 20892-9760, USA

**Keywords:** cervical cancer, mHealth, implementation research, evaluation, Tanzania

## Abstract

Introduction: The World Health Organization (WHO) recommends visual inspection with acetic acid (VIA) for cervical cancer screening (CCS) in lower-resource settings; however, quality varies widely, and it is difficult to maintain a well-trained cadre of providers. The Smartphone-Enhanced Visual Inspection with Acetic acid (SEVIA) program was designed to offer secure sharing of cervical images and real-time supportive supervision to health care workers, in order to improve the quality and accuracy of visual assessment of the cervix for treatment. The purpose of this evaluation was to document early learnings from patients, providers, and higher-level program stakeholders, on barriers and enablers to program implementation. Methods: From 9 September to 8 December 2016, observational activities and open-ended interviews were conducted with image reviewers (n = 5), providers (n = 17), community mobilizers (n = 14), patients (n = 21), supervisors (n = 4) and implementation partners (n = 5) involved with SEVIA. Sixty-six interviews were conducted at 14 facilities, in all five of the program regions Results SEVIA was found to be a highly regarded tool for the enhancement of CCS services in Northern Tanzania. Acceptability, adoption, appropriateness, feasibility, and coverage of the intervention were highly recognized. It appeared to be an effective means of improving good clinical practice among providers and fit seamlessly into existing roles and processes. Barriers to implementation included network connectivity issues, and community misconceptions and the adoption of CCS more generally. Conclusions: SEVIA is a practical and feasible mobile health intervention and tool that is easily integrated into the National CCS program to enhance the quality of care.

## 1. Introduction

Sub-Saharan Africa has a cervical cancer incidence rate of 50.9 cases per 100,000 women, the highest incidence rate in the world [1]. The burden of disease is disproportionately high in low- and middle-income countries (LMIC), which account for 85% of cases and nearly 90% of cervical cancer deaths worldwide [2]. In Tanzania, cervical cancer is the most common cause of death from cancer for women [3]. Sporadic uptake cervical cancer screening (CCS), prevalence of high-risk oncogenic human papillomavirus (HPV) subtypes, and relatively high rates of human immunodeficiency virus (HIV) co-infection compound risks of infection [4]. Survival rates are low, with more than half of women diagnosed in Tanzania dying of the disease, as they receive their diagnosis at an advanced stage when curable treatment options are limited [5]. In 2015, 82% of women in Kilimanjaro, Tanzania reported they had knowledge of cervical cancer, although only 6% had ever been screened [6].

It is well documented that in order to avoid progression to later stages of disease, effective prevention strategies must be employed widely and routinely. Screening and treatment for pre-cancer of the cervix is a secondary prevention strategy used globally in many low-resource health care settings to prevent cervical cancer. In these contexts, the main method of secondary prevention for cervical cancer is visual inspection of the cervix with the naked eye after the application of 3% to 5% acetic acid solution. Visual inspection with acetic acid (VIA) is regarded as the best approach in most low and some middle-income income countries and has been endorsed by the World Health Organization (WHO) when combined with ablative ‘see and treat’ approaches at the time that a woman receives VIA [7,8]. VIA can be performed by nurses and other skilled health practitioners and is inexpensive and non-invasive. It can be practiced broadly in lower-level health facilities as well as in HIV care and treatment programs, as part of regular screening services for HIV-related malignancies. More importantly, VIA provides instant results, and those eligible for treatment can receive treatment with ablative methods (thermal ablation or cryotherapy) on the very same day in the same health facility. This “see and treat” method promotes adherence to treatment, as it is offered immediately after diagnosis, thus minimizing the likelihood of loss to follow-up associated with patient referrals for treatment at an alternate facility or higher level of care [9]. Since 2020, the WHO has recommended HPV DNA testing as a primary screening method in preference to cytology and visual inspection with acetic acid (VIA). The absence of carcinogenic HPV types indicates an extremely low immediate risk of precancer/cancer and a reassuring low risk of cervical cancer for longer subsequent periods [10]. The superior sensitivity of HPV testing as compared to cytology and VIA and the reassurance against cervical cancer following a negative HPV test result is increasingly leading to its adoption as the main primary screening method in many countries worldwide [10]. VIA still plays a role as a triage step among HPV-positive women, but efforts to improve its performance are needed. In fact, as the interpretation of VIA is highly subjective, quality control has proven to be highly variable, impacting profoundly on both sensitivity and specificity [11,12,13]. A triage strategy is important to reduce the number of women referred to colposcopic biopsy in search of precancer or treatment. In many settings, the HPV prevalence is too high and the healthcare capacity too low to refer all HPV-positive women to colposcopic biopsy or to treat all.

Informed by the 2020 WHO cervical cancer prevention guidelines, Tanzania’s national cervical cancer prevention program (CECAP) is piloting the use of HPV DNA testing as a primary screening tool across four regional sites in 2024. Additional research programs [14] have been implemented in several sites to evaluate effective strategies for larger-scale implementation of HPV DNA testing through self-sampling in health facilities and in rural and urban communities. At present, and while further research to inform the implementation and scale-up of HPV DNA testing as a primary screening strategy is completed, the CECAP program continues to deliver screening services that are underpinned by the previous WHO guidelines with the use of VIA and the “see and treat” method and utilizes nurses and other clinicians providing frontline, non-specialist care. The program consists of six days of competency-based training, and continued mentorship by experienced senior CCS trainers to complete screening with VIA in the field under supervision, as well as routine follow-ups. Due to resource constraints, available training staff, and geographic logistics, follow-up training and supportive supervision opportunities do not always occur, and maintaining competency and quality of screening among CCS providers in the CECAP program has been a notable challenge. There has been a significant decline in skill retention and VIA quality, as a result of the lack of post-training mentorship and continuous technical support for CCS providers, combined with the frequent turnover of trained staff [15,16].

Digital cervicography has been known to improve the quality of VIA [17,18]. It uses a digital camera to transmit an image of the cervix to a screen/monitor, using higher clarity and resolution to review the image, than is possible with the naked eye [17,18]. Images are transmitted to experts at coordinating sites for review [19,20,21]. However, the program requires particular infrastructure and equipment, making it less feasible in many low-resource settings, including Tanzania. During the study period, a digital cervicography program was piloted at a single site in northern Tanzania but did not expand to other locations due to resource constraints. However, the growing prevalence of smartphones and dependable mobile networks across Tanzania presented a hopeful resolution to the challenges encountered during the program’s attempted scaling. Program developers believed that smartphone cameras could provide a feasible avenue to enhance supervision, oversight, and measurable quality assurance, especially in remote areas [15].

The Smartphone-Enhanced Visual Inspection with Acetic acid (SEVIA) program included the dissemination of smartphones to CCS providers, together with a smartphone application permitting real-time, secure sharing of de-identified VIA cervical images and relevant clinical information by health providers to expert ‘reviewers’. Image reviewers are senior VIA providers (i.e., gynecologists, and other skilled physician/non-physician VIA trainers within Tanzania’s CECAP program who review images in real-time, providing supervision and mentorship to nurses and non-physician clinicians [15]. When there is discordance between the health provider and the reviewer, the application sends a notification, allowing the reviewer to provide secure in-app feedback and recommendations on the diagnosis and treatment plan. This is meant to ensure the most accurate diagnosis and treatment plan for the client, while also promoting continued learning and high-quality supervision for the provider [15].

The program efficacy of the SEVIA concept was tested in a pilot study in the Kilimanjaro region from June 2014–March 2015 [16] and transitioned to scale in a pre-post study to evaluate the effectiveness of the intervention at existing CECAP sites using VIA. The program was delivered intensively for 6 months at 24 health facilities, followed by a 6-month maintenance phase. This comprehensive implementation evaluation was completed to evaluate client and health provider-related implementation outcomes, and reports on experiences, perspectives, and general acceptance of the program by both groups.

## 2. Methods

The study was conducted from September to December 2016 in 5 regions of Northern Tanzania (Kilimanjaro, Arusha, Kigoma, Bukoba, and Tanga). Measures included a thorough evaluation completed during the implementation phase, in order to understand barriers and enablers of the intervention, and early learnings from program consumers, providers, and higher-level stakeholders. The primary author, holding an MPH, trained in qualitative research epidemiology, and possessing Swahili language skills, completed observational activities and open-ended interviews with image reviewers (n= 5), providers (n = 17), community mobilizers (n = 14), patients (n = 21), supervisors (n = 4) and implementation partners (n = 5) involved with the SEVIA program. Facilities were selected for assessment based on proportional calculations of regional representation in the program, and by inclusion of rural/urban and higher/lower performing facilities. Image reviewers, providers, supervisors, and implementation partners were selected by consecutive sampling, and contacted in advance by telephone. Patients and community mobilizers were sampled by convenience sampling on-site. A total of 66 interviews were conducted with program informants at 14 facilities, in all 5 of the program regions. Additional interviews conducted at program sites run by an international nongovernmental organization were excluded from this analysis, as these clinics functioned on a private for-profit model, and the patient population was of a higher socioeconomic status.

The Reach, Efficacy, Adoption, Implementation, and Maintenance (RE-AIM) Framework [22,23] and the Consolidated Framework for Implementation Research (CFIR) [24,25] are implementation science frameworks that were developed to guide systematic assessment of multilevel implementation contexts and assist in translating research to evidence-based practice. An implementation evaluation framework was designed by adopting similar implementation outcome variables [26]. Implementation outcome measures included acceptability, adoption, appropriateness, feasibility, fidelity, implementation costs, coverage, and sustainability. Data collection was performed iteratively, with continual refinement of the protocol. A relationship was established prior to study commencement. Participants were aware of the research team’s goals in performing the research. There was no presence of non-participants.

One female researcher conducted, transcribed, and coded all participant comments and interviews, allowing for data immersion and obtaining an overall sense of the data. Interviews were conducted in-person at the sites, with a few implementation partner interviews being conducted over the phone. Most interviews lasted 20–30 min. Repeat interviews were not carried out. The questions, prompts, and guides were provided by the study authors. Field notes were made during and after the interviews were conducted. Transcripts were not returned to participants for comments and/or corrections. Using the outcome variable framework [26], content inductive analysis was used for each variable [27]. An open coding approach was adopted, forming a general description of the research topic by generating categories and subcategories as they emerged [27]. This systematic approach was appropriate for open-ended interviews to determine trends and patterns. Discussion with a second researcher familiar with the data confirmed emerging categories. Lastly, to minimize confirmation bias, two external female researchers, trained in qualitative research methods, who were unfamiliar with the data and preliminary findings, independently coded all participant interviews inductively with QSR-NVivo [28]. Secondary researchers undertook a thematic analysis, established consensus on emerging themes, and resolved conflicts to determine final results once data saturation was met. Findings were confirmed with the preliminary researcher, and all three collaborated to determine the final themes presented. Participants did not provide feedback on the findings.

## 3. Results

A full extrapolation of results by implementation outcome is provided in Table 1. Overall, our findings indicated that the intervention was trusted by both patients and providers, was easily implemented within regular/routine practice and care and assisted in both the quality of diagnosis/care offered to patients, and in the oversight and ongoing training of providers. Women appreciated seeing an image of their cervix as it provided an immediate understanding of their health and body, and providers expressed increased confidence in the care they were able to provide. Community-level knowledge or trust in CCS in general was seen as a barrier impacting the overall acceptability of the program’s desired outcome. The most common misconceptions of screening were that the procedure would be intrusive and painful/uncomfortable, and that the reproductive parts would be removed for examination. Women additionally expressed a reluctance to screen for fear of receiving a positive result or prognosis. The SEVIA application was seen to be quickly understood by providers, but issues related to mobile network coverage impacted the speed and regularity with which providers could share images with reviewers in real-time. While the intervention appeared to reach both women of middle and lower socioeconomic statuses, it was difficult to determine the true reach of the program, as interviews were generally conducted with respondents who had already reached the centre. Community mobilizers did provide some useful information in regard to coverage/reach of the program, but their position was inherently biased as they were tasked with mobilizing within a particular community. The extent to which more rural communities not encompassed within the study had an understanding of screening or cervical cancer in general could not be established. Motivations for participation were unclear, as providers and image reviewers were provided small per diems to see patients and review images. In the absence of compensatory incentives in the long term, program sustainability could not be sufficiently determined.

## 4. Discussion

It is critical to acknowledge the amount of time that has passed since data collection. These data on cervical cancer were collected between 2014–2016; however, it retains significant relevance in 2024, offering critical insights into the disease’s impact and the efficacy of interventions. Despite efforts to mitigate challenges such as screening and treatment deficiencies, cultural barriers, and limitations in healthcare infrastructure, cervical cancer incidence and mortality rates have exhibited minimal change, highlighting its persistent public health importance in Tanzania. This study enables the evaluation of preventive strategies and identification of current gaps, and emphasizes the ongoing imperative for sustained global and local efforts to address disparities in healthcare access and outcomes. Furthermore, these data serve as a pivotal reference for ongoing initiatives aimed at reducing the burden of cervical cancer and enhancing women’s health in Tanzania and globally.

A number of key findings were highlighted across all settings and by numerous respondent types. A major challenge was network connectivity issues and interruptions especially in rural health facility settings, which was discussed by providers, reviewers, and other program stakeholders. While Tanzania has seen significant improvements in the speed and reliability of mobile telephone networks in recent years, coverage issues persist in many, primarily rural areas. Certain cellular providers have strong networks in particular geographic regions but not in others. As the program spans the country, program implementers were challenged to select a carrier to provide reliable coverage at all sites, and some sites struggled more than others. A more comprehensive coverage plan with multi-providers may be required to ensure every facility is equipped with a phone with sufficient network coverage. This would dramatically decrease the necessity of providers saving images, reduce any delays of image reviewers responding to providers, and allow the intervention to function in real-time as intended.

An overwhelming majority of providers and reviewers reported that SEVIA integrated seamlessly into their general practice of screening, giving it a high feasibility and probability of long-term integration in standard practice. However, it was unclear whether study participants would be equally motivated to continue using SEVIA once study per diems ceased. A significant portion of the clients who were screened during the intervention phase of the study were acquired at outreach campaigns (small mobilization efforts conducted at facilities in rural villages) where providers were given per diems. Campaigns were funded by the research study in order for newly trained providers to reach their minimum number of screens for certification, as well as to increase the number of study participants observed in a short period of time. It was later observed that a consistently low number of women reported to facilities for screening after the intervention phase of the program.

The intervention was also found to increase the knowledge and skills of providers with limited training. SEVIA permitted providers to better visualize the cervix, and consequently increased confidence in their diagnosis and role as screeners in general. These findings are supported by other mHealth interventions in resource-limited countries which demonstrate the value of mobile phones in tackling barriers to service provision and improving both the range and quality of services offered by community-level health providers [29,30,31]. For example, in a qualitative study evaluating the acceptability and usability of a mobile phone-based ophthalmic testing system to perform comprehensive eye examinations in Nakuru, Kenya, healthcare providers reported that the tool aided them in detection and diagnosis, provided decision support, improved communication among providers, and assisted in education and training [31,32].

Improved health education and enhanced health literacy are essential for SEVIA or similar mobile health programs and/or applications supporting cervical cancer screening services to be broadly accepted and sustainably implemented at the community level, especially in rural and remote communities. While there appeared to be widespread acceptability for the use of the smartphone to capture images of the cervix by women who had already agreed to screen, there appeared to be low acceptability at the community level of cervical cancer screening in general. For example, a number of patients and providers commented there was a misconception that the reproductive parts would be removed for examination during the procedure. Others expressed fears that screening would be painful and uncomfortable, and feared receiving a positive result (especially among HIV+ women). As CCS is paired with HIV testing in Tanzania, this may especially deter women most at risk of receiving a positive HIV diagnosis. Additionally, results indicated that the majority of women were not aware that early-stage detection and treatment by ablative methods could be performed on-site. These beliefs are aligned with other studies reporting barriers to cervical cancer screening uptake in the region. For example, a qualitative cross-sectional study in Lilongwe, Malawi, found barriers to CCS with VIA uptake to include misconceptions of screening procedures and fatalistic views on cancer in general [33]. In this study, most participants reported that prior to undergoing cervical cancer screening they had limited understanding of the process. Myths and misconceptions of the screening process included expectations that the exam would be painful, fear of receiving a positive screening result, distrust in healthcare workers and suspicion of specimen collection and removal of the uterus [33]. In another cross-sectional study assessing factors associated with cervical screening uptake among HIV-infected women at Mildmay, Uganda, where CCS with VIA was integrated into HIV care, respondents reported similar misconceptions related to screening, such as removal of their ovaries and/or uterus and “cutting off of flesh” [34]. While our study did not specifically seek to ascertain the acceptability of CCS in the general sense, these findings have implications for the potential impact SEVIA on target populations in the future. The distinct lack of knowledge about cervical cancer at the community level and primarily at rural community sites where our evaluation took place signals the need for more widespread education on cervical cancer, screening, and treatment. Our findings, in addition to others [11] demonstrate that much of the fear, mistrust, and misconceptions can be alleviated with targeted health education. Specifically, increasing awareness about the importance of regular screening, dispelling myths and misconceptions about cervical cancer and the HPV vaccine, and providing culturally sensitive information will reduce fear and mistrust. Results suggest that in order for SEVIA to reverse the trajectory of cancer diagnoses in the country, efforts must also be placed on community education, increasing health literacy, and general promotion of CCS. Educational initiatives should be delivered through community outreach, integration into school curricula, and leveraging media campaigns to ensure widespread dissemination.

Patients generally expressed positive experiences with SEVIA and appreciated the addition of smartphone technology to screening. The data suggest SEVIA may even serve to empower patients with respect to health education and individual/personal health literacy. Being able to directly visualize one’s own cervix and any associated lesions provides immediate reassurance and information about one’s health, thereby providing an opportunity for improved individual health literacy and understanding of self. Increased health literacy has been shown to improve one’s knowledge and self-care behaviours among individuals with various health conditions across socioeconomic and cultural settings [35,36,37]. Desire to know one’s health status was reported as a key outcome in the same study of VIA clients in Lilongwe noted above, “I did that [screening] because I wanted to know the condition of my body, you can just be staying and never be certain you are okay or not. So, this time I thought it wise to go get screened” [33].

As Tanzania moves towards more widespread implementation of HPV DNA testing as a primary screening strategy that can provide more broad population coverage (e.g., women can perform a vaginal self-swab in a rural community with the guidance of a trained community health worker and results can be quickly delivered back to a local screening nurse to communicate with the woman about the test result and next steps for follow-up screening and directing the woman to a local screening site for visual assessment for treatment if they are HPV DNA positive. The role of a mobile health platform such as SEVIA has the potential to support the rapid roll out and scale-up of new strategies that utilize HPV self-sampling.

As this study employed a cross-sectional design, temporality cannot be inferred. Additionally, given the challenges of sampling hard-to-reach populations, the present results from patients could underrepresent more marginalized women. All of the patient testimonies were obtained from women who reached health facilities in the program and agreed to be screened, thus a great deal of information is missing on the challenges of increasing uptake of CCS more generally. Finally, the variables in this analysis by all respondent types were self-reported and thus may be subject to social desirability bias.

## 5. Conclusions

Findings from this semi-structured qualitative implementation evaluation indicated that the SEVIA program was a highly regarded tool for the enhancement of CCS services in Northern Tanzania. Acceptability, adoption, appropriateness, and feasibility of the intervention were highly recognized. It proved an effective means of improving good clinical practice among providers and fit seamlessly into existing roles and processes. While technical restraints caused adaptations to the intervention protocol, these are to be expected in the preliminary development of a technology. Alterations to the app were ongoing at the time of the study, and the usability of the tool was increasing. Network connectivity issues were a persistent challenge to program adherence in a number of locations and will need to be overcome for the program to be effective in the future. Allocation of permanent funds for outreach activities and more comprehensive community education and mobilization approaches are recommended, in order to increase the regularity of screening in general and access the most vulnerable women. In 2018, as a consequence of SEVIA supporting the development of good clinical practice among CCS providers through oversight and continuous training, Tanzania’s CECAP program integrated the SEVIA model into their Cervical Cancer Prevention Strategic Plan for 2020 to 2024 [37]. Funding restrictions and resource shortages have remained intermittent challenges, as has the COVID-19 pandemic, with health system resources re-directed to public health. At the time of writing, the SEVIA program was undergoing further scale-up and evaluation of a new version of the SEVIA mobile app that includes integration of HPV DNA test results and additional tracking functions and follow-up indicators to reduce loss to follow-up and support navigation of women to improved linkage to follow-up screening services.

## Figures and Tables

**Table 1 ijerph-21-00878-t001:** Results by Implementation Outcome Variable.

Implementation Outcome	Working Definition	Related Terms	Results
Acceptability	The perception among stakeholders (for example patients, providers, managers, policy makers) that the intervention is agreeable	Comfort, relative advantage, credibility	Trust was highlighted by mobilizers, providers, and image reviewers, as a key requirement for buy-in from the community. Respondents reported general trust among the community towards providers and emphasized the use of trusted community leaders as mobilizers. Knowledge that reviewers were specialists was also seen to instill trust among patients. For those who remained untrusting, dominant beliefs included: (1) that the reproductive parts would be removed during the screening process for examination; (2) the intimate/intrusive nature of the procedure; (3) fear of pain from speculum; (3) fear of receiving a positive result (especially for HIV+ women); and (4) uncomfortableness with male providers. While some women feared screening services and were initially hesitant, they were quite willing to attend screening services once educated by community mobilizers. For example, one patient indicated that they were “scared at first, but once provided with health education felt totally fine.” Most women did not appear to have preliminary knowledge of cervical cancer or available screening services until program education was offered, and many were unaware that pre-cancerous treatment was available on-site. SEVIA campaigns (concentrated outreach activities) appeared to have increased awareness of cervical cancer, in addition to screening services available in the community. At a few sites, it was noted that clients may have preferred to receive screening services from a provider that they did not know (i.e., a foreigner or someone from the referral hospital) while others did not indicate a preference. Acceptability of the use of smartphones was widespread when proper pre-counselling was provided, and the cervix was shown post-procedure.
Adoption	The intention or action to carry out the program	Uptake, utilization, intention to try	Program staff appeared highly motivated to implement the smartphone-based and mobile App-supported program. For providers in particular, adoption was high, as the smartphone was an add-on to their existing VIA practices, and a tool that simplified their roles. One provider stated, “The addition of the phone has simplified my work because I can see the cervix from a different angle”. While other program staff (image reviewers, community mobilizers, supervisors, etc.) also appeared motivated to employ SEVIA, it was unclear whether this motivation was driven by true readiness to adopt the intervention, or by implementation phase compensation. For example, one mobilizer explained that their motivation to continue their work was that “many women need help”, while another mobilizer indicated that they had concerns about a “gap in funding” which would influence their ability to continue with program implementation.
Appropriateness	The perceived fit of the program in the setting	Relevance, perceived fit, compatibility, perceived usefulness or suitability	Appropriateness was deemed very high in all program regions, and by all stakeholders involved. All providers indicated that they liked using the technology. Reasons included: (1) taking a picture of the cervix with the smartphone allowed them to visualize it better and make a more accurate assessment which enhanced confidence in their role as a provider; (2) they appreciated having another specialist available to corroborate their diagnosis; and (3) they were receiving critical ongoing training/education from their image reviewer. Both image reviewers and providers noted that the addition of the smartphone did not interfere with their existing processes or outstanding responsibilities. However, a desire for ongoing mentorship in the form of refresher training was consistently recommended among providers, mobilizers, and image reviewers. Despite many providers feeling unsatisfied with the duration of training and/or lack of refresher training, the majority still reported feeling comfortable training others. The majority of patients indicated they had an overall positive experience with SEVIA and were comfortable with the addition of the smartphone to the screening process. Reasons included: (1) they liked being able to see a picture of their cervix after the screening—it helped provide them reassurance about their health status; and (2) they liked that the provider could double check the diagnosis with a specialist.
Feasibility	The practicality of the program being carried out in the setting	Practicality, fit, utility, trialability	While many providers noted challenges in learning the technology in the early stages of implementation, all expressed mastery of the application within 4–5 months of using it. Overall, SEVIA appeared to assist providers in their roles, and helped streamline processes rather than creating additional work. Image reviewers did not report difficulty learning the technology or express any challenges in reviewing images on top of their pre-existing responsibilities. Apart from the addition of the smartphone, all program resources were covered in existing VIA programs, although these resources did not account for an increase in patients due to SEVIA efforts. This may be a barrier to program implementation in settings where many women are being screened, or where supply chain issues are a challenge. Network connectivity was seen as the biggest challenge at almost all of the facilities visited, which created barriers to implementation in many settings. Other challenges included sharing phones among providers at a given site, and delays in reviewer responses. The former speaks to a challenge of limited program resources (i.e., phones) and the latter due to network connectivity issues.
Fidelity	the degree to which the program is carried out as intentionally planned	Adherence, delivery as intended, integrity, quality of programme delivery, intensity of dosage of delivery	A number of adaptions were made throughout the lifecycle of the program, most notably to the application. While the program initially intended for providers to send patient files and receive a response from reviewers within a 5-min timeframe, this was not the case in all instances. In many cases, files were being saved within the application, and sent when network connectivity returned. Protocol for follow-up differed by facility (the patient returns for results in person, or is called if treatment is required), but in all cases, loss to follow-up was not seen as a concern (all women returned or were easily contacted by phone). While it was intended that all providers offer extensive group education sessions to clients, one-on-one counseling on the purpose of the phone being used and explanation of where the image is being sent, as well as showing the woman a picture of her cervix, it was noted that not all these practices were being employed by providers in every setting and there was limited oversight to ensure adherence to desired standards.
Implementation costs	the incremental costs of carrying out the program	Marginal cost, total cost	Initial implementation costs were higher during the implementation stage, which included training expenses, purchasing of phones, and mass screening campaign expenses, however, once this phase was completed, program maintenance fees were quite low.
Reach	The ability of intervention to reach target population/s	Coverage, range, accessibility	In most regions and program sites, respondents reported that women of all economic positions were accessing screening services, but more so among middle- and lower-income women. In many of the rural settings, transport issues were noted as a barrier, especially for women of the lowest income bracket. It was also noted that women of the highest income bracket may have been receiving screening elsewhere (i.e., private facilities), or may not have recognized the need to access screening services altogether. For example, a patient from the Kilimanjaro region noted the perception that, “higher class go to town, while the lower class go to the village health care centres”. In terms of geographic accessibility, the majority of respondents discussed inadequate program reach to rural areas where screening is more difficult to access and expressed the need for increased mobilization to villages and rural areas. For example, one patient expressed there was a need for “more education to women in the villages: in the interior. They don’t get information, so it is not easy to convince them to come [for screening].”
Sustainability	The ability of the program to continue independent of research implementation	Longevity, maintainability, support	Sufficient infrastructure existed to sustain the program’s long-term-reliable technology, widespread adoption, motivated stakeholders, and alignment with the local policy climate.

## Data Availability

The original contributions presented in the study are included in the article. Further inquiries can be directed to the corresponding author.

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
