# Peer review of "An Implementation Evaluation of the Smartphone-Enhanced Visual Inspection with Acetic Acid (SEVIA) Program for Cervical Cancer Prevention in Urban and Rural Tanzania"

_ijerph, 2024, doi:10.3390/ijerph21070878_

Round 1

Reviewer 1 Report

Comments and Suggestions for Authors

Dear Authors,

Topic is very interesting. I have a few comments:

1. Abstract - too long.

2. Introduction: 

- data is old - for 2020 why you didn't use some fresh data?! (line 47)

- then when writing about Tanzania you used data for what year ? - from the references [2] [3] it looks that you took data for 2018 and 2010; so it means that data applying to world and then to Tanzania are not comparable however you jointed them together - you wrote about cases in Tanzania as percentage of world data while data for world is from 2020 and data for Tanzania for 2010 and 2018; 

- it is difficult to find the formulation of article purpose in the introduction. 

3. Method: 

- you made research for 2016 (line 146) you should provide some statistics for Tanzania and all word for 2016 in the introduction apart from some fresh comparable data 

- the rest is well explained also in the Appendix (it means - model); 

4. Results are clearly presented.

5. Discussion - it would be good to indicate / provide exactly what kind of health education (line 286) should be implemented. 

6. Limitations and Conclusions are fine. 

7. References - there are mistake as there are two number 1. 

Reviewer 2 Report

Comments and Suggestions for Authors

The authors present an implementation evaluation of SEVIA program in Tanzania. The manuscript is understandable, informative, the methods are clearly presented, the results are systematically shown, the discussion is of high quality and conclusions follow the results. The language is understandable and no editing is needed as far as I can assess as a non-native speaker. I am only curious why the manuscript was send for publication almost 8 years after the research and program started. It would be useful if authors would explain this somewhere in the manuscript. 

Round 2

Reviewer 1 Report

Comments and Suggestions for Authors

Dear Authors,

corrections improved the understanding of article and improved the quality of the paper.